# Muscle Activation in Traditional and Experimental Barbell Bench Press Exercise: A Potential New Tool for Fitness Maintenance

**DOI:** 10.3390/sports7100224

**Published:** 2019-10-17

**Authors:** Andrea Melani, Giuliana Gobbi, Daniela Galli, Cecilia Carubbi, Elena Masselli, Luca Maria Neri, Gaspare Giovinco, Antonio Cicchella, Laura Galuppo, Valentina Presta, Mauro Vaccarezza, Marco Vitale, Prisco Mirandola

**Affiliations:** 1PhD Program in “Systems, Technologies and Devices for Movement and Health” at the University of Cassino and Southern Lazio, 03043 Cassino, Italy; andreamelani@gmail.com; 2Sport and Exercise Medicine Centre (SEM), Department of Medicine and Surgery, University of Parma, 43125 Parma, Italy; giuliana.gobbi@unipr.it (G.G.); marco.vitale@unipr.it (M.V.); prisco.mirandola@unipr.it (P.M.); 3Unit SBiBiT (Biomedical, Biotecnological and Translational Sciences), Department of Medicine and Surgery, University of Parma, 43125 Parma, Italy; cecilia.carubbi@unipr.it (C.C.); elena.masselli@unipr.it (E.M.); 4Anatomy and Histology Section, Department of Morphology and Experimental Medicine and Surgery, University of Ferrara, 44121 Ferrara, Italy; Lumane265@gmail.com; 5Department of Civil and Mechanical Engineering, University of Cassino and Southern Lazio, 03043 Cassino, Italy; andya8380@gmail.com; 6Department for Life Quality Studies, University of Bologna, 40126 Bologna, Italy; antonio.cicchella@unibo.it; 7PhD Program in “Molecular Medicine” at the University of Parma, 43125 Parma, Italy; laura.galuppo@studenti.unipr.it (L.G.); valentina.presta@unipr.it (V.P.); 8Department of Human, Social and Health Sciences, University of Cassino and Southern Lazio, 03043 Cassino, Italy; mauro.vaccarezza@curtin.edu.au; 9School of Pharmacy and Biomedical Sciences, Faculty of Health Sciences, Curtin University, Bentley, Perth 6102, Western Australia, Australia

**Keywords:** bench press, inter-handle distance barbell, fitness maintenance

## Abstract

Background: The bench press exercise (BP) is commonly practiced in both recreational and professional training. The weight is lowered from a position where the elbows are at a 90° angle at the start and <90° at the end of eccentric phase, and then returned to the elbows extended position. In order to focus the exercise more on the triceps brachii (TB) rather than the pectoralis major (PM), the inter-handle distance (IHD) is decreased diminishing the involvement of the PM in favor of the TB. Purpose: To improve performance of the exercise by reducing force dissociation and transmitting 100% of the external load to the muscle tissue we propose a prototype of the barbell with a bar on which two sleeves are capable of sliding. The dynamic modifications of the IHD keep the elbow flexion angle constant at 90°. Results: Analysis of the inter-handle distance (IHD) signals of the upper body muscles showed a marked increase in muscle activity using the experimental barbell for the PM (19.5%) and for the biceps brachii (173%). Conclusions: The experimental barbell increased the muscle activity typical of the bench press exercise, obtaining the same training induction with a lower load and consequently preventing articular stress.

## 1. Introduction

The bench press exercise (BP) is widely used among both recreational and professional athletes for strength training. [1]. BP exercise requires balance to lift a weight that consists of a steel bar with circular cast iron discs that are fixed to the bar by means of collars or circular springs. The present-day barbells have all the same features as simple-to-use tools. Depending on the use of the tool (home fitness, fitness, Olympic activities), the length of the bar varies from 150 to 220 cm. The 220 cm bar is used for official Olympic lifting competitions and training activities. The diameter of the bar may vary from 25 to 30 mm, while the bushings at the ends which are designed to accommodate the overload may have the same diameter, or, as in the more professional use of barbell arms, a diameter of 50 mm. 

At the beginning of the exercise the subject is generally in a supine position on a bench under the barbell. The subject removes the barbell from the metal support gripping it with both hands so that the elbows form an angle of 90° with the bar, and the hands are in line with the elbows (Figure 1 panel A).

Then, the barbell is slowly lowered to the chest (Figure 1, panel B). After a pause, the barbell is returned to the starting position. The classic technique used by body builders and fitness practitioners is to use an inter-handle distance (IHD), so that at the start (when the barbell is positioned on its supports, Figure 1, panel A) the hands are in line with the elbows. This means that in the upper dead point (Figure 1, panel C) and lower dead point (Figure 1, panel B) the forearm is not perpendicular to the barbell [1]

During ascending or descending movements, the forearm axis, will assume angles that are always different from perpendicularity. The direction of the force exerted from the elbow to the barbell forms corners of <90° (Figure 1, panel C), that results, necessarily, in significantly higher mechanical dissipation and thus less generated muscle power in relation to physical effort [1]. 

Some authors have analyzed the BP exercise through kinematic and electromyographic data acquisition [2]. The bench press is a multi-articular exercise and it has been demonstrated that the distance between the hands changes the involvement of the affected muscle groups [3,4,5]. The activations and the muscular synergies are correlated with: the subject’s experience [6], the use of a free barbell bar, or the assistance of a Smith machine [7], or with the use of a "peck deck" [8], and finally with the type of support (stable bench vs. unstable surfaces) [9].

The muscle groups involved in the bench press are the pectoralis major (PM), the elbow extensors represented by the triceps brachii (TB), the anterior deltoid (AD), the serratus anterior muscle (GD), and the elbow flexors represented by the biceps brachialis (BB), as well as the peri-articular shoulder muscles group [3,4,5]. By focusing exclusively on the TB and the PM which are the muscles most involved during the exercise, a decrease in the inter-handle distance (IHD) diminishes the involvement of the PM in favor of the TB [5].

Moreover, in a traditional barbell, the constriction caused by the handles restricts movement of the PM muscle reducing its range of motion at short and long muscle lengths. On the contrary, if the subject uses a shorter IHD (corresponding to the biacromial distance) the PM would be able to shorten completely, but the exercise would lose its effectiveness because it would predominantly recruit the elbow extensor muscles and not the PM [3].

Several handles that allow different ranges of motion and speed can be applied during BP performance [10,11]. However, little is known about the effects of these on the activity of the muscles, elbow joint kinetics, muscle fatigue, and exercise variation in training involved in BP [12]. 

Although there are chest press exercises with convergent movement, they do not show the typical feature of the barbell as a tool commonly used by the sports population that can stimulate all the shoulder stabilizing musculature and movement control mechanisms. Here we show the prototype of a barbell, consisting of a bar on which two sleeves (which act as grips) are capable of sliding in opposite directions to keep the center of mass of the barbell equidistant from each hand. This is achieved by having the two handles bound to each other so that the displacement of one will cause the displacement of the other symmetrically with respect to the center of the barbell. 

The starting hypothesis is that the new barbell increases the muscle activity typical of the bench press exercise regardless of the increase in external load on the bar, thus obtaining the same training induction with a lower load and consequently less articular stress. 

The new concept (free-grip) barbell is here compared with the traditional (locked) barbell using electromyography (EMG).

The barbell under study requires the development and learning of a specific execution technique, so that during the concentric phase the participants will approach the grips by making them converge toward the center of the barbell, while in the eccentric phase the grips diverge. This movement will keep the grips constantly aligned vertically with the elbow, allowing both an overload optimization and a complete muscular excursion from maximum lengthening to maximum shortening.

As the large pectoral muscles are the main target muscles of the exercise, we chose to use a horizontal bench [3,4] for the tests, and a distance between the grips not less than 200% of the biacromial distance [3]. Figure 2 (please insert here) provides a picture of the exercise execution with the free-grip barbell

To familiarize the volunteers with the free-grip barbell, all subjects were asked to perform familiarization lifts (four series) and, after a break of at least 5 min, a test with the traditional barbell. Each test included a minimum of 20 repetitions: the order of the tests with the locked or free-grip barbell was randomized. With the aim of working with two comparable systems, instead of a traditional barbell, we used a free-grip barbell with a blocked sliding mechanism and the weight load was matched between conditions. The bar weighs 24 Kg. While recognizing that different percentages of maximal force output could describe different muscle activation patterns, it was not considered necessary to perform the exercises with a specific percentage of the maximum, since the goal was to compare muscle activations at the same weight and not according to a percentage of the maximal force output [5]. At the end of the test with the locked barbell, the EMG signal was obtained for a maximum voluntary isometric contraction (MVIC) expressed in an intermediate position between full extension and full flexion. During all trials, especially in MVICs, subjects were stimulated verbally with the aim of obtaining a maximum value. The collected EMG data were analyzed using Microsoft Excel software (Excel 2011 for Mac, Microsoft, Albuquerque, NM, USA) while kinematic data were analyzed through Kinovea software (Version 0.8.15, Kinovea, https://www.kinovea.org). To obtain good experimental repeatability subjects were asked to perform each test three times with a break of at least 5 min between trials to avoid a fatigue effect. For each subject, the average value of the three tests and their standard deviations were calculated. 

## 2. Participants and Methods

### 2.1. Participants

Human volunteers who participated in the experiment (two females and seven males aged 22 to 49 years—average 29.9 years) practiced recreational physical activity, but not regularly. All procedures met the 1964 Helsinki declaration and its later amendments or comparable ethical standards. Data, reported in this manuscript, were anonymously generated. Subjects were exempt from the study if they reported any possible physical, psychological, or social injury. The procedure was approved by Comitato Etico Area Vasta Emilia Nord (prot. 33584 del 05/09/2018). 

Exclusion criteria were: (1) reduced muscular strength (cut-off for muscular strength was to perform the exercise with the barbell alone without additional overloads), (2) absence of fluid execution of the technique, and (3) professional bodybuilder activity. All participants were free of upper body orthopedic injuries or limitations.

### 2.2. Description of the Device

The device is shown in Figure 3. 

Two rectangular (4 × 3 mm^2^) deep channels were carved out of a 30 mm rod. The terminal part of the rod was then drilled 6 mm wide and 32 mm long for the depth of the entire rod diameter. At the end of the same rod, a 6 × 32 mm^2^ milling was carried out for the entire diameter of the rod. The first hole houses a steel cable of 2 mm, while the second receive the axis of a 29 mm diameter pulley on which the cable itself may rebound to reverse its direction from one burst to another. The grips, in addition to presenting a tooth able to slide in the guide preventing rotation, have, at their end, an ancho point for a cable which, once run through the rod by its length in the direction of one end, is wrapped around the pulley and then travels along the guide channel in the opposite direction, to pass under the same handle and end its run by fastening to the second handle.

## 3. Electromyography (EMG)

During test execution the electromyographic signals of the pectoralis major surface, anterior deltoid, brachial triceps, and brachial biceps were detected through the 4-channel EMG-2X VD12 electromyograph (SATEM, Rome, Italy) of the Satem. Electrodes were placed on the skin after shaving, with abrasion according to surface EMG recommendations for non-invasive muscle evaluation [13]. Specifically, pre-gelatinized bipolar electrodes of 9 mm diameter were placed 20 mm apart oriented longitudinally along the tested muscle fibers. Mass poles have been placed on two electrically neutral points (sternum and olecranic beak). The analog signal was processed with a 20 to 1000 Hz second order bandwidth filter, and a 50 Hz −35db notch (following manufacturer’s instructions) filter to isolate it from any network noise, and then straightened and rectified at a constant of 20 ms. Sampling took place at a frequency of 1/10 s.

Integral of the curve was then calculated as the sum of the individual values sampled by the EMG. The value of the area under the EMG curves was then divided by the number of contractions to obtain a mean value of contraction (VMC) for each muscle tested. The difference between the EMG signal of the traditional barbell and the free-grip barbell was expressed as the percentage of the value of the barbell with the highest value. In this way, absolute values have been reduced to an index that allows comparison regardless of the individual maximum strength, maximum voluntary contraction (MVC) [14,15]. In this regard, the literature suggests that raw EMG data should be normalized through the values of MVC. However, it should be noted that this procedure is only useful when the EMG data to be compared has been collected on different days (i.e. with repositioned electrodes) or when the data to be compared are from different subjects or different muscle groups.

## 4. Statistics

Electromyography data were normalized as a percentage of MVIC. Statistical tests were repeated after normalization through MVIC. Results of the EMG data for muscle groups were analyzed for distribution and to distinguish the data that followed normal distribution, a Kolomogorov–Smirnov test was applied. Statistical significance was assessed through Student’s t tests for normal distribution phenomena and Wilcoxon for other phenomena. The alpha level for significance was set at *p* < 0.05. Effect size was calculated to 0.2 for 9 participants.

## 5. Results

With the aim of comparing two types of barbells for optimizing muscle activation in bench press exercise, we employed electromyography (EMG). By overlaying the graphs of the two exercises (with locked and free-grip barbell) it was possible to select an identical analysis window for the two exercises. It is important to emphasize that, since the two bar systems are similar but not identical, EMG peaks may not correspond to identical angular moments. Therefore, to synchronize the peaks we performed a shift on the axis of the abscissas (Figure 4).

The comparison of the graphs was calculated over a period of seven concentric-eccentric contraction cycles. The length of the time was chosen by observing the graphs, excluding the queues and the regions of the tracks that showed obvious heterogeneity in muscle activation patterns. By assuming the free-grip barbell test as the test with the highest intensity signal, the difference between the two test signals was calculated. 

Since bench press mainly affects the PM muscle [3,4] it was expected that the test reassessment variability index was low especially for this group. The other muscles tested are affected in a variable way depending on the personal style that each of the subjects use when lifting the barbell. Statistical analysis confirmed this hypothesis.

In Figure 5 the EMG mean values for each examined subject are reported.

White and black bars refer to locked barbell (TB) and free-grip barbell (EB), respectively. The statistical analysis showed that pectoralis major, anterior deltoid and biceps brachii rejected the normal distribution hypothesis. On the contrary, normal distribution was verified for the triceps brachii. Therefore, different tests were applied. In detail, the Wilcoxon test was applied to the pectoralis, deltoid and triceps, while the Student’s t test was applied to the biceps. A statistically significant increase in muscle activity using the free-grip barbell for the PM (19.5%) and the BB (173%) with *p* < 0.01 was observed. Group mean values were the following: PM with locked barbell (6646.74 ± 1738.07); PM with free-grip barbell (9056.51 ± 740.62); AD with locked barbell (6791.02 ± 8708.87); AD with free-grip barbell (6840.23 ± 6503.47); BT with locked barbell (3932.57 ± 1901.41); BT with free-grip barbell (4226.38 ± 1339.26); BB with locked barbell (1855.24 ± 972.98); BB with free-grip barbell (4991.33 ± 8440.59). However, group mean values were not considered for the results since the distributions (except for the triceps brachii) were not significant.

These results are consistent with a greater involvement for the PM caused by less weight overload thanks to the constant arm-balance incidence angle and a greater muscular excursion for the same time that translates into a more efficient transfer of load; for the BB a greater commitment is due to the strength needed to overcome the friction of the slipping of the handles. 

## 6. Discussion

The bench press exercise (BP) is commonly practiced both in recreational and professional training to increase muscle strength and induce muscle hypertrophy. There is no age limit to practicing BP providing it is performed in adapted physical activity programs. Aging is associated with a progressive decrease of muscle mass and function, with a consequent decline of muscle strength and physical performance of elderly people [16]. Strength training with BP could be useful for preventing sarcopenia and maintaining muscle strength and function reducing articular stress.

We here studied the prototype of a barbell that optimizes strength training by increasing power production and reducing tendon stress. Our results evidences that a statistically significant increase in pectoralis major and biceps brachii activity was observed with a free-grip barbell compared to a locked barbell. Some authors could observe that similar results in peak of muscle activity could be expected with dumbbell presses [7,17,18]. However, we could expect greater activation time in the free-grip barbell bench press than in the dumbbell bench press. In fact, Welsch, Bird and Mayhew (2005) [17] showed that the barbell bench press determined greater activation time both in the pectoralis major and in anterior deltoid, with respect to dumbbell bench press. Moreover, the free-grip barbell bench press would present lower stability requirements. Of note Saeterbakken, van den Tillaar and Fimland (2011) [18] demonstrated that the dumbbell bench press has higher stability requirements than the barbell bench press.

Finally, another difference between the free-grip barbell and dumbells could be the resistance caused by the sliding of the handles (an effect that is not present in the use of dumbells) and the possibility of creating a different ceiling (greater) by using a barbell instead of two free dumbbells.

Thus, although our results were obtained in a sample of young people, they could be likely generalized and applied to conditions that require stability of the joints in older populations. In fact, muscle power training has been shown to have clear advantages over traditional strength training for improving functional performance in older adults. The prototype of the barbell presents friction issues that become evident to subjects when an overload of 10 kg is added to the barbell. For this reason, the tests were performed with the unload balance and without the calculation of 1RM in accordance with what has already been done for other similar studies [5]. At this intensity the influence of friction becomes negligible and analysis of the problem has shown that friction is generated by the handles and the anti-rotation device. During the tests, both problems were resolved by requiring subjects not to rotate the handle, applying a silicone lubricant and, above all, selecting subjects whose strength and technique allow for the correct use of the tool.

In this study the data were collected on the same day, a few minutes apart, and compared the same muscle groups of the same subject. Statistical tests were performed on absolute values and after normalization through MVC. In fact, the repeatability of the EMG tests carried out on the same day has been questioned in some studies [14,15,16,19,20]. However, it is considered without reasonable doubt that the transformation of the difference between the two moments in a percentage index is more reliable than normalization made through an MVC at an unspecified angle of isometry. 

It is conceivable that the use of imprecise technique tends to reduce the PM’s load more than the tool can. On the other hand, the barbell, like all the tools and exercises in which a mechanical guide is absent, requires a technique that maximizes the effects of training. 

Another important limitation of the study is the small size of the sample and the lack of elderly people on our preliminary study population. However, it should be noticed that the sample was homogeneous and statistical tests were adapted for a small sample. Further research should include an increase in the sample size and older individuals. Although large and unresolved areas of research remain, in this manuscript we evaluated a new conceptual barbell with the aim of: (1) Establishing clear functions and definitions of BP muscle stabilizers; (2) Studying muscles with the greatest potential to overcome sticking regions; and (3) Determining changes in muscle activity and performance.

## 7. Perspectives

The use of bench press exercise is widely diffused both in recreational and professional training. Here, we studied the prototype of a barbell (free grip) that optimizes strength training by increasing power production and reducing tendon stress. Likely, this type of barbell can be a valid alternative to the traditional one. However, this prototype is unable to meet the need to increase overload beyond the limits imposed by the subject. For this reason, a second prototype is being studied (from the operation already provided in the same patent field) which involves rolling the grips through the interface of rollers or ball bearings. The anti-rolling device of the handles, essential to avoid the cables guiding the movement, must be reconsidered as it has turned out to be responsible for some fluid leakage. To circumvent this phenomenon there are two possibilities: the use of ball bearings embedded in semicircular milling developed in the longitudinal direction of the bar (as shown in the Appendix A), or the use of a non-circular bar coupled with the use of roller bearings.

## 8. Practical application

The proposed device aims to increase muscle loading and power output without enhancing the external load during the bench press exercise, employing a biomechanical geometric arrangement of two handles applied to the barbell. This can be helpful in everyday practice to maximize the training effect and to reduce elbow joint loads.

## Figures and Tables

**Figure 1 sports-07-00224-f001:**
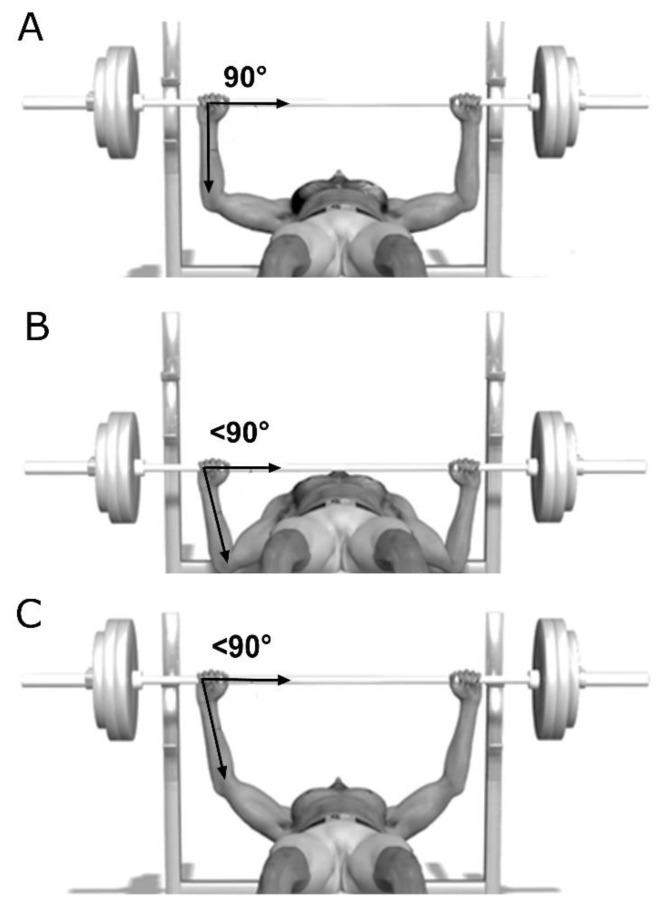
Bench press dynamic. Schematic representation of angles formed by the elbows and barbells during the performance of a traditional bench press exercise. The angle of the line hand–elbow and the barbell is reported. When the barbell is positioned on its support the angle is 90° (panel **A**). The start and end position are shown in panels **B** and **C**, respectively.

**Figure 2 sports-07-00224-f002:**
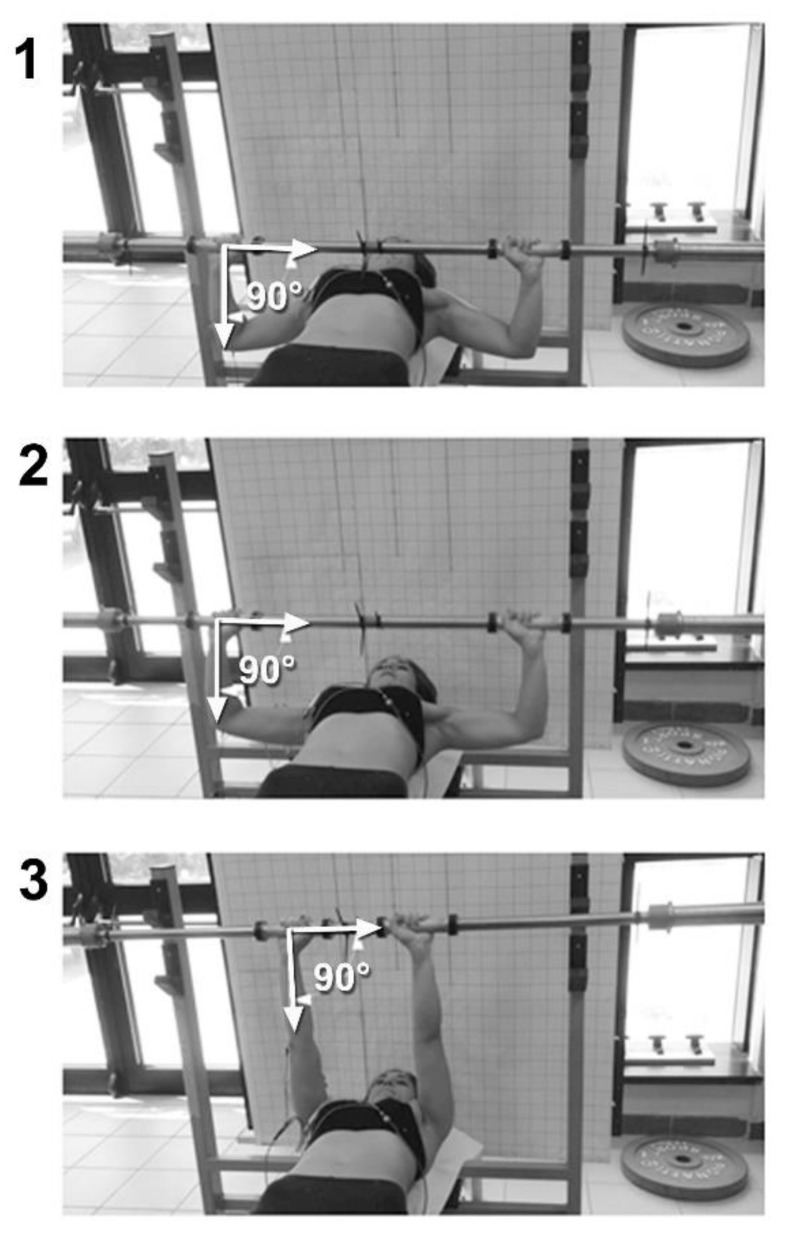
Execution with the free-grip barbell. The grips are constantly aligned with the vertical to the elbow (90° angle), allowing both an overload optimization and a complete muscular excursion from maximum elongation (panel **1**) to maximum shortening (panel **3**), an intermediate position between the maximum elongation and the maximum shortening (panel **2**).

**Figure 3 sports-07-00224-f003:**
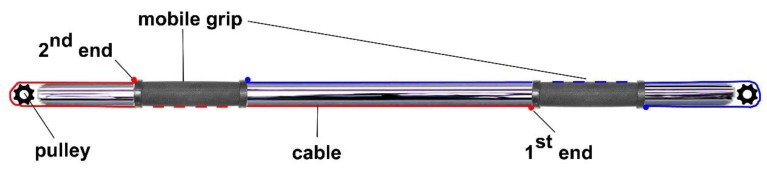
Schematic representation of the device. The handle (mobile grip) is tied to a cable (1^st^ end, red line) that runs in the opposite direction through a pulley engaging (2^nd^ end) the contralateral grip. A second cable (blue line) is engaged to the handles and run on the pulley at the opposite end of the barbell.

**Figure 4 sports-07-00224-f004:**
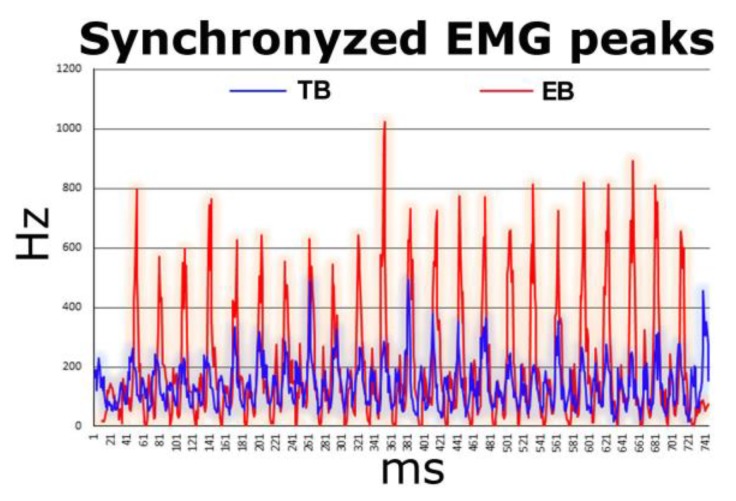
Electromyography **(**EMG) test performed both with the locked barbell (TB) (blue line), and the free grip-barbell (EB) (red line). Numeric EMG data was plotted on line-graphs by superimposing the two-to-two trials so that the test with the traditional barbell (blue line) and test with the experimental barbell (red line) would be on the same chart. A shift of the x axis data (time) was performed to synchronize the EMG peaks of the first test with the peaks of the second. An exemplificative result is reported.

**Figure 5 sports-07-00224-f005:**
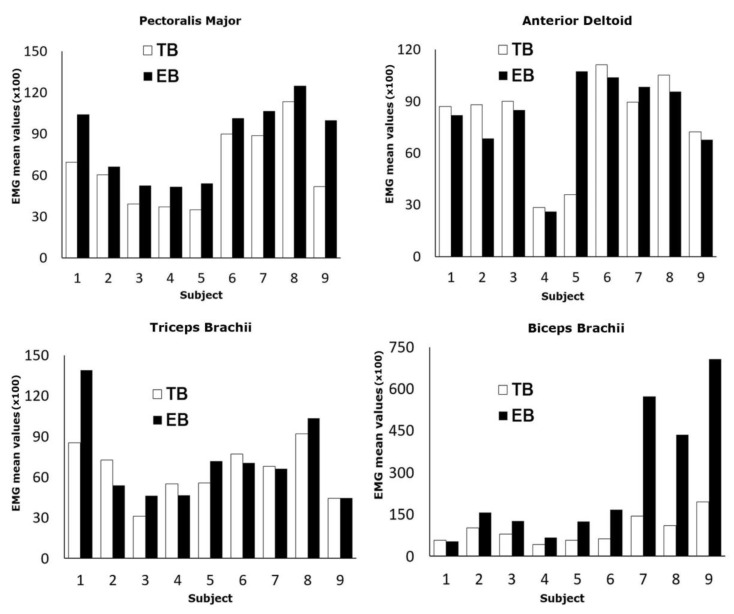
EMG analysis of upper body muscles during bench press training. EMG test were performed both with the locked barbell (TB) and the free-grip barbell (EB). Electromyography of the pectoralis major (PM), anterior deltoid front bands (AD), triceps brachii (TB), and biceps brachii (BB) is shown. Mean values of EMG data and EMG normalized through the values of maximum voluntary reduction (MCV) are reported. For PM EMG the critical value of W is 0, for N = 8; *p* ≤ 0.01. For BB EMG the W-value is 1, for N = 9; *p* ≤ 0.01. For DA EMG the W-value is 11, for N = 7; *p* > 0.05. For TB EGF the t-value is 0.643164; *p* > 0.53.

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
