# Peer review of "Muscle Activation in Traditional and Experimental Barbell Bench Press Exercise: A Potential New Tool for Fitness Maintenance"

_sports, 2019, doi:10.3390/sports7100224_

Round 1

Reviewer 1 Report

The paper needs extensive rework. Please look at the grammar throughout the paper and in addition make sure that claims in the introduction are backed up with references. This is missing in a number of the paragraphs. Please change the word "rocker" to "barbell" for ease of reading and perhaps change the acronym for the pectorals to more traditional verbage along with make sure that all acronyms used in the abstract are introduced before being used without context for them. 

Author Response

The paper needs extensive rework. Please look at the grammar throughout the paper and in addition make sure that claims in the introduction are backed up with references. This is missing in a number of the paragraphs. Please change the word "rocker" to "barbell" for ease of reading and perhaps change the acronym for the pectorals to more traditional verbage along with make sure that all acronyms used in the abstract are introduced before being used without context for them. 

R: We thank the reviewer. We checked grammar and references throughout the manuscript we changed the word from rocker to barbell, we changed the acronym for pectoral and we controlled the context of acronyms.

Reviewer 2 Report

Review sports-599868– “Muscle activation in locked and free grip bench press exercise: a potential new tool for fitness maintenance in all ages

The manuscript explores the EMG differences in a two different bench press techniques in nine men and women. The first technique is a traditional barbell bench press and the second involves an experimental barbell that allows the hands to slide closer together during the upward movement. The experimental barbell increased pectoralis major and biceps brachii EMG activity.

General Comments

The concepts in the manuscript are very interesting. However, the information about the rocker needs to be explained in detail early in the manuscript (introduction) to help the reader understand the study. Additionally, the writing needs to be clearer throughout the paper. For example, the authors use a number of different terms to describe the two experimental conditions traditional barbell (locked grip) vs experimental barbell (new concept rocker, free grip). Lastly, the common English terms should be used throughout the manuscript (e.g. anterior deltoid not deltoid front band and biceps brachii not brachial biceps).

Minor Comments.

Title

The title should be a more clear description of what is in the manuscript. I don’t think you can use the term “all ages”. Also using the term traditional barbell and experimental barbell may be more clear than locked and free grip.

 Abstract

Please include the purpose and/or hypothesis in the abstract.

Lines 39. I think you mean 19.5% not 19,5%

Line 40. Muscle stress does not seem like the appropriate term.

Need to define BB and EMG before you use these terms in your abstract.

 Introduction

Line 47-48. The second sentence in the introduction seems unnecessary. 

Methods

Line 107. I think you mean 29.9 years not 29,9 years.

Line 107. I am not sure what amatorial physical activity is?

Lines 112. How did you screen for bench press technique. Also what was your cut-off for muscular strength?

Figure 3 should be moved up earlier in the manuscript when the authors describe the rocker in the introduction.

Which condition was first? Was this randomized for each participant?

Results

Mean EMG results (of all participants) need to be included for each muscle group measured. (This would be nice in a table).

Lines 209. I think you mean 19.5% not 19,5%.

Lines 211-214. This should be in the discussion.

Is there any demographic information that can be included? Ages, height, weight? Also how much weight did each participant lift? How much weightlifting experience do the participants have? (how many years? And how many times a week?)

Discussion

The discussion should begin with a summary of the results.

The authors need to clearly explain in the discussion why this experimental barbell is optimal compared to the traditional. I understand that there is more EMG activity, but what is the benefit of the experimental barbell, since I could also add more weight to a traditional barbell to increase the EMG activity.

Line 254, 265. Power was not measured in this study. 

Author Response

Review sports-599868– “Muscle activation in locked and free grip bench press exercise: a potential new tool for fitness maintenance in all ages

The manuscript explores the EMG differences in a two different bench press techniques in nine men and women. The first technique is a traditional barbell bench press and the second involves an experimental barbell that allows the hands to slide closer together during the upward movement. The experimental barbell increased pectoralis major and biceps brachii EMG activity.

General Comments

The concepts in the manuscript are very interesting. However, the information about the rocker needs to be explained in detail early in the manuscript (introduction) to help the reader understand the study. Additionally, the writing needs to be clearer throughout the paper. For example, the authors use a number of different terms to describe the two experimental conditions traditional barbell (locked grip) vs experimental barbell (new concept rocker, free grip). Lastly, the common English terms should be used throughout the manuscript (e.g. anterior deltoid not deltoid front band and biceps brachii not brachial biceps).

R: We thank the reviewer. We add description of the rocker in the introduction. We changed the name of the two experimental conditions to “locked grip barbell” (instead of traditional barbell) and “free grip barbell” (instead of experimental barbell) throughout the manuscript. Moreover we now used common english terms for muscles like anterior deltoid and biceps brachii.

Title

The title should be a more clear description of what is in the manuscript. I don’t think you can use the term “all ages”. Also using the term traditional barbell and experimental barbell may be more clear than locked and free grip.

R: we thank the reviewer. We modify the title as suggested. The title now is “Muscle activation in traditional barbell and experimental barbell bench press exercise: a potential new tool for fitness maintenance 

Abstract

Please include the purpose and/or hypothesis in the abstract. We thank the reviewer. We modify the abstract as suggested

Lines 39. I think you mean 19.5% not 19,5%.

R: Yes, we apologise for the mistake.

Line 40. Muscle stress does not seem like the appropriate term.

R: We thank the reviewer. We changed muscular stress with muscle activity.

Need to define BB and EMG before you use these terms in your abstract.

R: We thank the reviewer. We introduced complete names before the acronyms. 

Introduction

Line 47-48. The second sentence in the introduction seems unnecessary.

R: We thank the reviewer and we removed the sentence. 

Methods

Line 107. I think you mean 29.9 years not 29,9 years.

R: we thank the reviewer and we apologise for the mistake.

Line 107. I am not sure what amatorial physical activity is?

R: We apologise with the reviewer. We changed the sentence to recreational physical activity.

Lines 112. How did you screen for bench press technique. Also what was your cut-off for muscular strength?

R: We thank the reviewer. Participants of this study practised this technique at recreational level. Cut-off for muscular strength was to perform the exercise with the barbell alone without additional overloads. We introduced the sentence in the manuscript, lines 140-141.

Figure 3 should be moved up earlier in the manuscript when the authors describe the rocker in the introduction.

R: We thank the reviewer. We moved up the figure 3 in the introduction and it has now become figure 2 

Which condition was first? Was this randomized for each participant?

R: Yes, it was randomised for each participant

Results

Mean EMG results (of all participants) need to be included for each muscle group measured. (This would be nice in a table).

R: we introduced mean values in the text (lines 212-218). However, since normal distribution was verified only for triceps brachii, we could not use mean values as results.

Lines 209. I think you mean 19.5% not 19,5%.

R: we thank the reviewer and we apologise for the mistake

Lines 211-214. This should be in the discussion.

R: We thank the review. We moved in the discussion

Is there any demographic information that can be included? Ages, height, weight?

R: No, considering the research design we decided not to collect them.

Also how much weight did each participant lift?

R: We thank the reviewer. The weight of the barbell is not important for the purposes of the test as the analysis of the data has been normalized also as a function of an isometric ceiling; in any case the balance weighs 24 Kg. We inserted the sentence lines 117-119.

How much weightlifting experience do the participants have? (how many years? And how many times a week?)

R: We thank the reviewer. The participants practised bench press at recreational level not regularly.

Discussion

The discussion should begin with a summary of the results.

R: we thank the reviewer.We add a sentence in the discussion to summarise results.

The authors need to clearly explain in the discussion why this experimental barbell is optimal compared to the traditional. I understand that there is more EMG activity, but what is the benefit of the experimental barbell, since I could also add more weight to a traditional barbell to increase the EMG activity.

R: We thank the reviewer. The proposed device aims to increase muscle loading and power output without enhancing the external load during the bench press exercise, employing a biomechanical geometric arrangement of two handles applied to the barbell. This can be helpful in everyday practice to maximize the training effect and to reduce elbow joint loads(lines 310-314).

Line 254, 265. Power was not measured in this study.

R: We thank the reviewer. We apologise and correct as suggested

Reviewer 3 Report

The authors test the use of an experimental barbell that allows the trainee to keep elbow flexion at 90 degrees throughout the bench press exercise by essentially having the grip of the barbell vary. THis is an innovative approach to change muscle activation patterns during the Bench Press.

Major comments:

Change abbreviation of Pectoralis Major to PM, this is more common in the literature. Can the authors elaborate on how the variable bench press used in their study differs from dumbbell presses? To the reviewer it seems like the freedom provided by this barbell is similar to that provided by dumbbells which also allows the trainee to bring the hands closer together during the end of the concentric phase. In particular, no differences in Pectoralis Major EMG activity has been reported using the dumbbell or barbell bench press, yet differences seem to occur in the present work with the variable barbell (Saeterbakken, van den Tillaar, & Fimland, 2011; Schick et al., 2010; Welsch, Bird, & Mayhew, 2005). Rocker is used throughout the manuscript to denote the barbell, I would use the word barbell and state that you used a free grip vs. locked barbell.

Line specific comments:

Line 31: Consider revising description to tstate that elbow is at 90 degrees angle at the start and is below 90 at the end of the eccentric phase. “Straight to the finish” is confusing as a way to denounce the start position. Line 39: Should be TB ins Line 85: The statement “GP fibers will only be moderately shortened” requires a reference. Line 106: Change amatorial to regular (provided that is what meant) and define regular (X amount of years of resistance exercise) Line 112: How was knowledge of the BP assessed? Line 114: What is meat by absence of professional bodybuilder activity? Professional bodybuilders were excluded from the study. Line 161: Report how many familiarizations were done by participants. Line 166: It is reported that the weight lifted was matched; however, the actual amount of weight lifted was not reported. Please report this. Line 186: Add ‘the’ before bench press. Line 204-214: Please also report mean group values (free grip vs. locked BP) for each muscle here instead of only showing the individual participant values. The group means can be reported in the text or table.

Literature

Saeterbakken, A. H., van den Tillaar, R., & Fimland, M. S. (2011). A comparison of muscle activity and 1-RM strength of three chest-press exercises with different stability requirements. J Sports Sci, 29(5), 533-538. doi:10.1080/02640414.2010.543916

Schick, E., Coburn, J., Brown, L., Judelson, D., Khamoui, A., Tran, T., & Uribe, B. (2010). A Comparison of Muscle Activation Between a Smith Machine and Free Weight Bench Press. J Strength Cond Res, 24, 779-784. doi:10.1519/JSC.0b013e3181cc2237

Welsch, E. A., Bird, M., & Mayhew, J. L. (2005). Electromyographic activity of the pectoralis major and anterior deltoid muscles during three upper-body lifts. J Strength Cond Res, 19(2), 449-452. doi:10.1519/14513.1

Author Response

The authors test the use of an experimental barbell that allows the trainee to keep elbow flexion at 90 degrees throughout the bench press exercise by essentially having the grip of the barbell vary. THis is an innovative approach to change muscle activation patterns during the Bench Press.

Major comments:

Change abbreviation of Pectoralis Major to PM, this is more common in the literature.

R: We thank the reviewer, we changed as suggested.

Can the authors elaborate on how the variable bench press used in their study differs from dumbbell presses? To the reviewer it seems like the freedom provided by this barbell is similar to that provided by dumbbells which also allows the trainee to bring the hands closer together during the end of the concentric phase. In particular, no differences in Pectoralis Major EMG activity has been reported using the dumbbell or barbell bench press, yet differences seem to occur in the present work with the variable barbell (Saeterbakken, van den Tillaar, & Fimland, 2011; Schick et al., 2010; Welsch, Bird, & Mayhew, 2005).

R: We thank the reviewer. We agree that, likely, the activation levels of Pectoralis Major could be similar with both presses. However, we could expect greater activation time in free grip barbell bench press than in dumbbell bench press. In fact, Welsch, Bird and Mayhew (2005) showed that barbell bench press determined greater activation time both in Pectoralis Major and in Anterior Deltoid, with respect to dumbbell bench press.

Moreover, free grip barbell bench press would present lower stability requirements. Of note Saeterbakken,  van den Tillaar and Fimland (2011) demonstrated that dumbbell bench press have higher stability requirements than barbell bench press.

Finally, another difference between free grip barbell and dumbells could be the resistance caused by the sliding of the handles (an effect that is not present in the use of dumbells) and the possibility of creating a different ceiling (greater) by using a rocker instead of two free dumbbells.

Rocker is used throughout the manuscript to denote the barbell, I would use the word barbell and state that you used a free grip vs. locked barbell.

We thank the reviewer. We modified as suggested.

Line specific comments:

Line 31: Consider revising description to tstate that elbow is at 90 degrees angle at the start and is below 90 at the end of the eccentric phase. “Straight to the finish” is confusing as a way to denounce the start position.

R: We thank the reviewer. We modified the sentence that now is:

“The weight is lowered from a position where the elbows are at 90°C angle at the start and <90°C at the end of eccentric phase where the bar touches the chest and returned to the elbows extended position”.

Line 39: Should be TB ins.

R: We thank the reviewer and correct as suggested.

Line 85: The statement “GP fibers will only be moderately shortened” requires a reference.

We thank the reviewer. The reference is Barnett, C., Kippers, V., & Turner, P. Effects of Variations of the Bench Press Exercise on the EMG Activity of Five Shoulder Muscles. The Journal of Strength & Conditioning Research 1995,9, 222-227.

Line 106: Change amatorial to regular (provided that is what meant) and define regular (X amount of years of resistance exercise)

R: We thank the reviewer. The participants practised bench press at recreational level not regularly.

Line 112: How was knowledge of the BP assessed?

R: We thank the reviewer. By performing 4 series of familiarization

Line 114: What is meat by absence of professional bodybuilder activity? Professional bodybuilders were excluded from the study.

R: Yes, professional bodybuilders were excluded from the study.

Line 161: Report how many familiarizations were done by participants.

R: We thank the reviewer. Participants performed 4 series of familiarization. We added a sentence in lines 114-115.

Line 166: It is reported that the weight lifted was matched; however, the actual amount of weight lifted was not reported. Please report this.

R: We thank the reviewer. The weight of the barbell is not important for the purposes of the test as the analysis of the data has been normalized also as a function of an isometric ceiling; in any case the balance weighs 24 Kg

Line 186: Add ‘the’ before bench press.

R: Yes, thank you.

Line 204-214: Please also report mean group values (free grip vs. locked BP) for each muscle here instead of only showing the individual participant values. The group means can be reported in the text or table.

R: we thank the reviewer and we add mean values in the text. Please find it below the text:

Group mean values were the following: PM with locked barbell (6646.744±1738.068); PM with free grip barbell (9056.511±740.623); AD with locked barbell (6791.022±8708.869); AD with free grip barbell (6840.233±6503.473); BT with locked barbell (3932.567±1901.41); BT with free grip barbell (4226.378±1339.26); BB with locked barbell (1855.244±972.979); BB with free grip barbell (4991.333±8440.592).

 However, since the distributions for all the muscles except biceps brachii was not normal, we did not use these values for analysis.

Literature

Saeterbakken, A. H., van den Tillaar, R., & Fimland, M. S. (2011). A comparison of muscle activity and 1-RM strength of three chest-press exercises with different stability requirements. J Sports Sci, 29(5), 533-538. doi:10.1080/02640414.2010.543916

Schick, E., Coburn, J., Brown, L., Judelson, D., Khamoui, A., Tran, T., & Uribe, B. (2010). A Comparison of Muscle Activation Between a Smith Machine and Free Weight Bench Press. J Strength Cond Res, 24, 779-784. doi:10.1519/JSC.0b013e3181cc2237

Welsch, E. A., Bird, M., & Mayhew, J. L. (2005). Electromyographic activity of the pectoralis major and anterior deltoid muscles during three upper-body lifts. J Strength Cond Res, 19(2), 449-452. doi:10.1519/14513.1

Round 2

Reviewer 1 Report

This is a great improvement, be sure to check your number of significant digits. Otherwise this looks ready to go.

Author Response

This is a great improvement, be sure to check your number of significant digits. Otherwise this looks ready to go.

R: We thank the reviewer for all suggestions. We checked significant digits and modified as follows: 

"Group mean values were the following: PM with locked barbell (6646.74±1738.07); PM with free grip barbell (9056.51±740.62); AD with locked barbell (6791.02±8708.87); AD with free grip barbell (6840.23±6503.47); BT with locked barbell (3932.57±1901.41); BT with free grip barbell (4226.38±1339.26); BB with locked barbell (1855.24±972.98); BB with free grip barbell (4991.33±8440.59)".

Reviewer 2 Report

The manuscript explores the EMG differences in a two different bench press techniques in nine men and women.  The first technique is a traditional barbell bench press and the second involves an experimental barbell that allows the hands to slide closer together during the upward movement.  The experimental barbell increased pectoralis major and biceps brachii EMG activity.

General Comments

This second version was much clearer to read.  However, please check the paper thoroughly there are still typos and inconsistencies.

Minor Comments. 

Abstract

I do not think you need so many short forms of words, i.e. TB for Triceps Brachii.  This makes it more difficult to read.

Line 31:  90o not 90o C

 Line 33:  Make sure it is clear that the 90o is with the hand on the bar and not at the elbow.

Introduction

Line 98:  It still says rocker bar.

Author Response

The manuscript explores the EMG differences in a two different bench press techniques in nine men and women.  The first technique is a traditional barbell bench press and the second involves an experimental barbell that allows the hands to slide closer together during the upward movement.  The experimental barbell increased pectoralis major and biceps brachii EMG activity.

General Comments

This second version was much clearer to read.  However, please check the paper thoroughly there are still typos and inconsistencies.

R: We thank the reviewer. We checked throughout all the manuscript and we removed all typos and inconsistencies.

Minor Comments. 

Abstract

I do not think you need so many short forms of words, i.e. TB for Triceps Brachii.  This makes it more difficult to read.

Line 31:  90o not 90o C

 Line 33:  Make sure it is clear that the 90o is with the hand on the bar and not at the elbow.

R: we thank the reviewer and we modified the abstract as suggested.

Introduction

Line 98:  It still says rocker bar.

R: We thank the reviewer. We corrected as suggested.